# Ribosomal DNA Abundance in the Patient’s Genome as a Feasible Marker in Differential Diagnostics of Autism and Childhood-Onset Schizophrenia

**DOI:** 10.3390/jpm12111796

**Published:** 2022-10-31

**Authors:** Elizaveta S. Ershova, Natalia N. Veiko, Svetlana G. Nikitina, Elena E. Balakireva, Andrey V. Martynov, Julia M. Chudakova, Galina V. Shmarina, Svetlana E. Kostyuk, Nataliya A. Salimova, Roman V. Veiko, Lev N. Porokhovnik, Aliy Yu Asanov, Vera L. Izhevskaia, Sergey I. Kutsev, Nataliya V. Simashkova, Svetlana V. Kostyuk

**Affiliations:** 1Research Centre for Medical Genetics, Department of Molecular Biology, 115522 Moscow, Russia; 2Mental Health Research Center, Kashirskoe Highway, 115522 Moscow, Russia; 3Department of Medical Genetics, I.M. Sechenov First Moscow State Medical University (Sechenov University), 115522 Moscow, Russia

**Keywords:** ASD, infantile autism, early-onset schizophrenia, childhood-onset schizophrenia, ribosomal genes, copy number

## Abstract

Introduction: Differential diagnostics of early-onset schizophrenia and autism spectrum disorders (ASD) are a problem of child psychiatry. The prognosis and relevant treatment are to a large degree determined by the correctness of diagnosis. We found earlier that leucocyte DNA of adult schizophrenia patients contained significantly larger copy numbers of ribosomal repeats (rDNA) coding for rRNA, than DNA of mentally healthy controls. Aim: To compare the contents of ribosomal repeats in the leucocyte DNA of children with schizophrenia, children with ASD, and healthy age-matched controls to estimate the possibility of using this genetic trait in the differential diagnostics of the two types of disorders. Patients and methods: Blood samples of patients with infantile autism (A—F84.0 according to ICD-10, N = 75) and with childhood-onset schizophrenia (SZ—F20.8 according to ICD-10, N = 43) were obtained from the Child Psychiatry Department of the Mental Health Research Center. The healthy control blood samples (HC, N = 86) were taken from the Research Centre for Medical Genetics collection. The recruitment of cases was based on the clinical psychopathologic approach. DNA was extracted from blood leukocytes with organic solvents. Nonradioactive quantitative hybridization technique was applied for determining the abundance of ribosomal repeats in the genomes. Statistical processing was performed using StatPlus, Statgraphics and MedCalc. Findings: DNA derived from SZ cases contained 565 ± 163 rDNA copies, which is significantly (*p* < 10^−6^) higher than the rDNA content in ASD cases (405 ± 109 copies) and controls (403 ± 86 copies). The HC and A groups did not differ by rDNA copy number (*p* > 0.4). The genetic trait “rDNA copy number in patient’s genome” can potentially be applied as an additional marker in differential diagnostics of childhood-onset schizophrenia and autism spectrum disorders.

## 1. Introduction

According to ICD-10, autism spectrum disorders (ASD) fall into the category F84 “pervasive developmental disorders”. Infantile autism is diagnosed based on developmental delays and abnormalities by the age of three and the occurrence of psychopathologic changes in social interaction, functions of communication, and behavior, which are restricted, stereotypic and monotonous (ICD-10, 1994 [1]). The features persist over a lifetime, but they may remain concealed until social requirements increase with the child’s age. The disturbances detected in childhood may become less visible in adult age in cases of adequately prescribed therapy and good management [2].

Childhood schizophrenia is related to the category F20.8 “other schizophrenia”. Childhood schizophrenia is “schizophrenia characterized by distinctness and polymorphism of the clinical picture with the onset in childhood. Also includes cases of schizophrenia arising in early childhood with a pronounced oligophrenic defect” [1].

Examination of the debut of suspected ASD and childhood-onset schizophrenia and their timely diagnosis are important challenges of child psychiatry [3]. However, the matter of differential diagnostics of infantile autism and childhood schizophrenia remains unsolved [4,5,6]. 

By the end of the 20th century, autism was fully separated from schizophrenia; moreover, the possibility that schizophrenia can debut in infancy was challenged and disputed [7]. Recent reports have suggested that autism spectrum disorders may be a specific preclinical phenotype for childhood schizophrenia [8]. The lack of agreement regarding the processes that take place during the disturbed child’s development also creates problems with classification and diagnosis. The timely differential diagnosing of infantile autism and childhood schizophrenia could reveal the disease at an early stage, reduce the rate of misdiagnoses, and consequently improve the quality of the patient’s life, due to adequate and consistent therapy. 

A search for biomarkers that could reliably differentiate between ASD and childhood schizophrenia is currently being carried out. A “Neuro-immuno-test” [9], hemostasiogram [10], electroencephalography, an assay of circulating DNA properties, and variability in the expression of genes for a series of signaling molecules [11,12,13] have been proposed.

Both ASD and schizophrenia are known to be etiologically underpinned by a combination of genetic and environmental factors [11,14,15,16,17]. Formation of an autistic or schizophrenic phenotype depends on many of genes, often shared by the autistic and schizophrenic pathways that regulate the development and function of the central nervous system. Genetic polymorphism studies have revealed numerous variants in hundreds of genes associated with ASD or schizophrenia. However, genetically diagnosing ASD or schizophrenia is complicated and not applicable in practice, despite the known genes and available techniques for genetic diagnostics.

Earlier, we revealed that genomes of adult schizophrenia patients contained more copies of genes for ribosomal RNA (rRNA) compared to genomes of mentally healthy subjects [18,19]. The human genome harbors some hundreds of copies of genes for rRNA encoding of three types of rRNA (18S, 5.8S, 28S rRNA) within an integrated unit termed ribosomal repeat, or rDNA. The ribosomal repeats are located on five pairs of acrocentric chromosomes in p-regions (short arms). Genes for 5S rRNA are located separately on chromosome 1 (Figure 1A). The p-regions of human acrocentric chromosomes can harbor zero up to several dozen or hundreds of rDNA copies. Each ribosomal repeat is ~43 kbp long and is composed of two parts: a transcribed region (a sequence 13,314 kbp long) and a non-transcribed intergenic spacer [20]. The transcription of rDNA is led by the RNA polymerase I enzyme. Proliferating cells have the most active rRNA production by polymerase I, and the amount of rRNA transcripts constitutes 35–60 percent of the total cellular RNA in proliferating cells [21]. The abundance of rDNA copies affects the total genomic expression level. The more copies of ribosomal repeats, the more ribosomes are produced, and the higher the rates of protein synthesis [22]. 

The fact of higher copy numbers of rDNA in the genomes of adult schizophrenia patients [18,19] suggests the potential feasibility of this genetic trait as a marker of differential diagnostics of childhood schizophrenia and autism. In order to verify, rDNA copy numbers should be determined in the genomes of children with autism. We could not find any early published reports of studies of rDNA abundance in autism. 

The aim of this study is comparative analysis of rDNA copy numbers in blood cell genomes in three groups of children: schizophrenia patients, autism patients, and healthy controls. 

## 2. Patients and Methods

### 2.1. Study Design

The total sample (N = 204) included three child cohorts: schizophrenia patients (N = 43), ASD patients (N = 75), and healthy children (N = 86). The patients were recruited from the Child Psychiatry Department of the Federal State Budgetary Scientific Institution “Mental Health Research Center” using the clinical psychopathologic approach. The approach included child observation in different situations with psychopathologic estimations of the behavior, emotional and cognitive manifestations, and social performance features. Additionally, patients’ medical records provided by parents were analyzed. DNA was isolated from two mL of the venous blood of cases (N = 118) and controls (N = 86). The nonradioactive quantitative hybridization (NQH, [18]) technique was used to determine rDNA copy numbers in 204 DNA samples from the three groups of children, and the data so obtained were used to compare the three groups by this value.

### 2.2. Ethics Expertise

Each child’s parent(s) signed an informed consent for venous blood sampling (2 mL) and the realization of scientific experiments with the biomaterial taken from the child. The study design was approved by the ethics committee of the Federal State Budgetary Scientific Institution “Research Centre for Medical Genetics” (protocol # 6/4 dated 15.11.2016).

### 2.3. Patients, Psychometric Scales, Inclusion/Exclusion Criteria

The study group included 118 patients with ASD and schizophrenia examined on the bases of clinical criteria of ICD-10 [1], DSM-5 [23], as well as age-matched healthy controls (86 healthy children). The age of patients was four to 12 years. Recruitment of the patients with ASD was carried out by a psychiatrist using the Childhood Autism Rating Scale (CARS) scoring [24]. Children with schizophrenia were estimated according to their PANSS scores [25]. The developmental dynamics was estimated using the Personal and Social Performance scale (PSP) adapted for childhood patients [26].

The inclusion criteria were: children with ASD with infantile psychosis and atypical infantile psychosis and children with early childhood-onset schizophrenia. The exclusion criteria were: syndromal autism cases (congenital metabolism defects, chromosomal abnormalities) and progressive degenerative diseases.

### 2.4. Clinical Analysis in the Subgroups

The group of ASD patients (A group) included children with autism of various severity diagnosed as “infantile psychosis” (F84.0) or “atypical infantile psychosis” (F84.1), Table 1. Catatonic disturbances occupied a central place in seizures. Motor excitation was accompanied by negativism. The patients demonstrated echopraxia, behavioral, and motor stereotypies. In periods of psychosis, speech development discontinued. During remission, ASD signs relieved, affection developed, and hyperkinetic disorders with attention deficit and psychopathic manifestations prevailed. Motor awkwardness was observed. 

The group of schizophrenia patients (SZ group) included children with a very early onset and a malignant and continued course of the disease (F20.8). The patients were characterized by violent manifestation with catatonic-regressive seizures at ages 1.5 to three years. A hallmark of the disease was rapid formation of negative changes: asthenia, fatigue, emotional exhaustion, regression of speech, and loss of motor and neatness skills. An oligophrenic defect formed during the manifest period and was retained for the lifetime of the patient. Approximately one third of the patients showed resistance to the antipsychotic therapy. The patients required permanent custody and care and lifelong psychopharmacotherapy. According to the social performance scale, the patients matched the third level of de-adaptation.

The healthy controls group (HC group) was formed from healthy age-matched children. The controls passed visual analysis of expert level EEG and comparative EEG-mapping of the brain to exclude any possible subclinical conditions.

### 2.5. Biochemical Techniques

#### 2.5.1. DNA Isolation and Quantification

Venous blood was drawn and placed in heparin tubes. DNA isolation was carried out using the standard extraction procedure with organic solvents, as described in detail earlier [18]. A solution composed of 0.04 M EDTA, 2% sodium lauryl sarcosinate and 150 μg/mL RNAase A (“Sigma”, USA) was added to the freshly drawn blood (45 min at 37 °C), followed by treatment with proteinase K (200 μg/mL, “Promega”, USA) for 24 h at 37 °C and extraction with equal volumes of a phenol/chloroform/isoamyl alcohol (25:24:1) mixture and a phenol and chloroform/isoamyl alcohol (24:1) mixture. DNA was then precipitated by adding 1/10 volume of 3 M sodium acetate (pH 5.2) and 2.5 volumes of ice ethanol. The DNA precipitate was collected by means of centrifugation at 10,000 G for 15 min at 4 °C, washed with 70% ethanol, dried and dissolved in water.

The concentration of DNA in the solution was determined using fluorescence with DNA-binding dye ‘PicoGreen’ (“Invitrogen”) and a tablet spectophotometer-fluorometer ‘EnSpire’ (“PerkinElmer”). The relative standard error of DNA quantification was two percent of the value measured.

#### 2.5.2. Determining rDNA Copy Count in DNA Samples

The NQH technique was described in detail in our previous report [18]. Denatured DNA was applied onto a filter (Optitran BA-S85, “GE Healthcare”) to make four dots for each sample. Standard samples of genomic DNA with a known rDNA content were applied onto the same filter to draw a calibration curve of the signal intensity versus rDNA copy number. Lambda phage DNA (50 ng/mL) was also applied onto the same filter to control the noise level. The DNA was immobilized by heating at 80 °C in vacuum for 1.5 h. After that, hybridization was carried out with a biotinylated DNA-probe, which contained a fragment of rDNA (“GenBank” #U13369, a region from −515 to 5321) cloned in pBR322 plasmid. After the hybridization had been finished, the membrane filter was treated with streptavidin conjugated to alkaline phosphatase (“Sigma”) and placed into a solution of substrates for alkaline phosphatase (BCIP /NBT), (Applichem). Afterwards, the filter was washed with water, dried in darkness and scanned. A special software, “Imager 6”, which calculated an integral signal intensity from each dot, was applied for quantification. Signals from all the dots that corresponded to the same sample were summed up, and mean and standard errors were calculated for each sample. The NQH method error was 5 ± 2%. The total method error for the rDNA copy number assay, which includes stages of DNA isolation and hybridization, was 11 ± 4%.

Unlike the generally acknowledged qPCR, the NQH technique requires a large amount of DNA (100 ng) and is not applicable to unique (single copy) genes. However, the NQH technique outputs a similar dependence of the signal intensity on the degree of DNA damage for both tandem rDNA and non-tandem repeats. Previously, it was shown that rDNA is significantly damaged under conditions of oxidative stress [18].

### 2.6. Statistical Analysis

The statistical data were processed with Excel Microsoft Office and StatPlus2007 (http://www.analystsoft.com, accessed on 20 October 2022) software packs. The sample comparison was based upon a null-hypothesis of the absence of difference between the two samples compared. The null-hypothesis was checked by the Mann–Whitney U-test. Distributions of copy numbers in the groups were compared using the Kolmogorov-Smirnov test (D and α) and the Kruskal-Wallis test (H and p). A difference was deemed significant in case of *p* < 0.001. The logistic curve was plotted using Statgraphics Centurion (www.statgraphics.com). ROC (Receiver Operating Characteristic) curves were drawn using MedCalc software (https://www.medcalc.org/manual/roc-curves.php, accessed on 20 October 2022). Each point on the curve displays a ratio between sensibility and specificity, which corresponds to a preset threshold value of rDNA copy number. The area under the curve (AUC) shows the difference between the groups analyzed.

## 3. Results and Discussion

The rDNA copy numbers were determined in 204 DNA samples isolated from the blood of three groups of children: ASD patients (A group), schizophrenia patients (SZ group) and healthy controls (HC). Experimental data showing rDNA copy numbers in the genome of each child are presented in Figure 1B. In Table 2, descriptive statistics data are listed.

In Figure 1B,C, the results of comparison of the groups of children by rDNA copy numbers (U-test) and comparison of the distributions of DNA samples by rDNA copy numbers (Kolmogorov-Smirnov test) in the groups are shown. For reference, dashed lines in Figure 1C display data for samples of healthy adult subjects (N = 814) and adult schizophrenia patients (N = 956), which were obtained and published earlier [18,19].

The genomes of children with diagnosed schizophrenia contained more ribosomal genes than those of healthy age-matched controls (*p* < 10^−7^) and children with diagnosed ASD (*p* < 10^−6^). Thirty percent DNA samples from the SZ group contained large numbers of rDNA repeats (>659 copies), which were not found in samples of the HC group. The SZ group had no subject with a rDNA copy number less than 300, while in the HC group, such samples constituted 17% (Figure 1B). The same result was obtained in our previous research when comparing rDNA copy numbers in healthy adult controls and adult schizophrenia patients [18,19].

The results obtained were quite predictable. Previously, using the model of replicative and natural aging, we showed that the number of rDNA copies in the human genome is a genetic trait that remains practically unchanged during the lifetime of one cultured cell pool or one human subject [27]. The exception is a small percentage of the genome containing hypermethylated rDNA copies localized outside the nucleolus structures. They are replicated with great difficulty. During replicative and, hypothetically natural aging, these “ballast” rDNA copies, which constitute up to 5–10% of the total number of rDNA copies in the genome, will be eliminated, resulting in a slight decrease in the total number of rDNA count. Genomes of adult patients with schizophrenia, who were found to have an increased number of rDNA copies, had harbored the same quantity or even slightly more copies of rDNA in childhood.

Thus, a very large number of rDNA copies in the genome of a child with a mental disorder may indicate an eventual schizophrenia case. Figure 1D presents the data of logistic regression and ROC analysis. The ROC curves (Figure 1D) suggest an estimated threshold value of the rDNA copy number to suspect the pathology when exceeded. The optimum ratio between sensitivity (real disease) and specificity (false positive diagnosis) is achieved at a value of 750 copies: 75% of cases are consistent with the diagnosis of schizophrenia and 25% are false positive. A very low rDNA copy number in a genome (less than 300) suggests a different mental disorder in the carrier.

The genomes of children with diagnosed ASD and those of healthy children did not differ by rDNA copy number. Autism is not associated with an increased number of rDNA copies in the patient’s genome. Only one child (1.3%) was found in the A group who had a rDNA copy number of 751, exceeding the threshold (the maximum value in the control group), whereas the SZ group included 14 (30%) patients with the same or a higher value. In this single case, a fact of misdiagnosing cannot be excluded.

In our previous articles, several times we reviewed various hypotheses to explain the fact of elevated rDNA abundance in the genomes of schizophrenia patients [18,19]. It became clear recently that the elevated rDNA abundance in the genome cannot be deemed a hallmark of schizophrenia when analyzing the total human population. An elevated rDNA copy number in the patient’s genome appeared to be typical for another, monogenic disease—cystic fibrosis, which is determined by mutations in the *CFTR* gene [28]. Another intriguing fact is that a large sample of long-livers contained no genomes with rDNA count of more than 500 copies, whereas the fraction of such high-copied genomes in subjects younger than 20 years amounted to 40% [27].

Analyzing these facts, we concluded that a large number of rDNA copies in the individual human genome is not the cause of the pathology. Most likely, this is a hallmark that the genome contains negative genetic components that require intensive protein synthesis during the embryogenesis; otherwise the embryo does not survive. Many rDNA copies can meet these embryogenesis needs, allowing a child to be born with a slight genetic pathology. However, the causal mutations that are contained in the genome will entail the disease in the future and shorten the lifespan.

Schizophrenia is a polygenic disease. The genomes of schizophrenia patients with many rDNA copies probably harbor certain mutations (rare variants) associated with the pathology. A very large number of rDNA copies in the genome is a reason for whole genome sequencing to identify mutations in the genes associated with the disease.

ASD is also classified as a polygenic disease. Apparently, genetic features in the genomes of autistic patients may also require intensive protein synthesis during embryogenesis to maintain homeostasis. Patients with autism do not differ from healthy children in terms of rDNA copy number in the genome. However, the need for an elevated level of protein synthesis can be met not only due to many rDNA copies in the genome (that provide for the synthesis of many ribosomes), but also to changes in the translation apparatus itself. Autism studies have discovered mutations in the genes that regulate the biogenesis and functioning of ribosomes, for example, in the genes for the Akt/mTOR/S6K1 [29] axis, translation initiation factor 4E (eIF4E) [30], and in the gene for translation repression factor FMRP [31]. These mutations boost the protein synthesis without changing the number of ribosomes and ribosomal genes.

The ribosomal repeat, which forms the nucleolus structure in the eukaryotic cell, performs another important function in addition to the canonic role it plays in ribosome biogenesis. The ribosomal repeats stabilize the heterochromatin of the entire nucleus, shifting the heterochromatin-euchromatin balance towards heterochromatin [32]. The larger the total size of the rDNA clusters, the more stable the heterochromatin and the less likely chromatin rearrangements occur in the nucleus. Perhaps, the large number of rDNA copies in schizophrenia makes it possible to block genetically determined chromatin instability and to help an embryo survive (to pass the zygotic selection).

## 4. Conclusions

Determination of the copy number of ribosomal repeats in the DNA isolated from blood cells of children with a mental disorder can potentially be used as a criterion of differential diagnostics of autism spectrum disorders and schizophrenia. A ribosomal repeat count of more than 750 in a child with a mental disorder would suggest diagnosing early-childhood schizophrenia, while a low count of 300 or less would exclude schizophrenia and point at ASD.

## Figures and Tables

**Figure 1 jpm-12-01796-f001:**
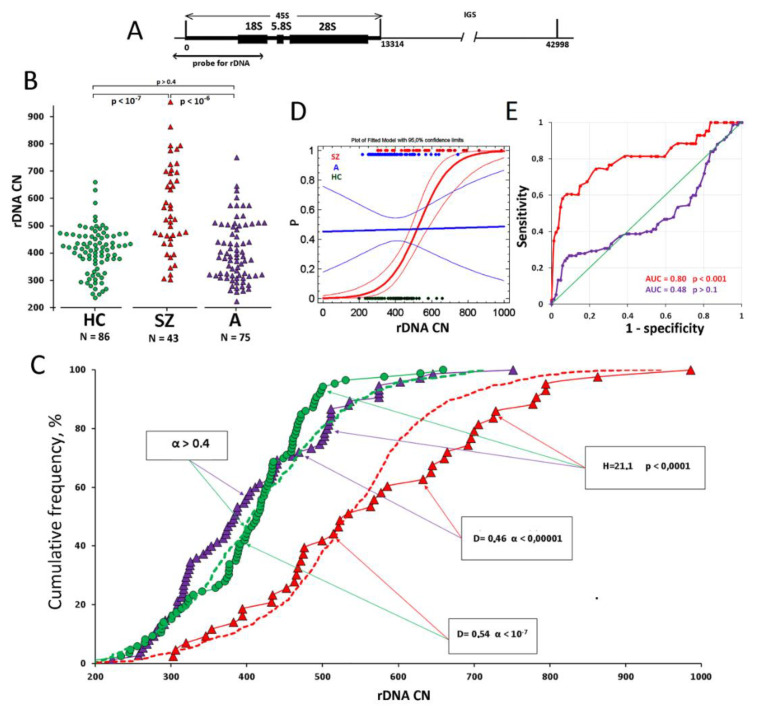
(**A**). Location of genes for rRNA on human chromosomes. Scheme of the ribosomal repeat unit. The area is shown that can be detected using a biotinylated DNA probe for rDNA. (**B**). The values of rDNA abundance obtained in the experiment for the three cohorts of children. (**C**). Comparison of the groups by rDNA CN distribution. The arrows indicate the groups compared, and the frames shows the statistical analysis data. (**D**,**E**). Logistic curves and ROC curves drawn for the groups of ill children compared to the controls.

**Table 1 jpm-12-01796-t001:** Clinic and demographic description of the study participants.

	A Group	SZ Group	HC Group
N	75	43	86
Age	6 ± 3	8 ± 3	7 ± 2
Gender, M/F (%)	58/17 (77%/23%)	34/9 (79%/21%)	56/30 (65%/35%)
Catatonic syndrome	The onset before the age of three. The phenomenon of sensory dissociation combined with catatonic/hyperdynamic disorders in the active period of the disease, replaced by psychopathic and affective symptoms by an age of 3–4. Duration of the active period of the disease is 1.5–2 years.	The onset in the crisis periods of 1.5 and 3 years. Hyper- and hypokinetic catatonia during the attack, motor stereotypies, neurosis-like disorders in remission. The duration of manifest psychosis is 2.5 to 4 years. A developed polymorphic psychotic attack at age points of 6–7 and 12–13 years.	Not observed
Delusions, hallucinations, elements of the Kandinsky syndrome	Not observed	Observed from an age of 4–5 during the attacks (not less than 1 month); in the remission, rudimentary deceptions of perception, unsystematic delusions of attitude, damage, poisoning	Not observed
Negative impairments	Delay or arrest of speech development. Mild/moderate autism.	Arrest of speech development or speech loss during manifest psychosis and incomplete recovery during remission. Incoherence of speech after the second psychotic attack. Short attention span. Concreteness and torpidity of thinking. Moderate/severe autism	Not observed
Cognitive impairments	Perverted type of cognitive dysontogenesis	Deficiency-progredient or defecting type of cognitive dysontogenesis	Not observed
Disease course and prognosis	Positive dynamics of the course of the disease with “practical recovery” in 10%, transition to “high functional” autism in 30%, regredient course in 60%. Psychopharmacotherapy exclusively during the active period of the course of disease, correction of emotional or psychopathic disorders in remission. Social rehabilitation.	Paroxysmal-progressive or continuous course. Bad prognosis in case of absence of timely prescribed psychopharmacotherapy. Frequent exacerbations with polymorphic symptoms. Nursing, long-term psychopharmacotherapy, socialization throughout lifetime.	Normal development

**Table 2 jpm-12-01796-t002:** Descriptive statistics for the rDNA copy number values in the three groups studied.

	Schizophrenia Patients (SZ Group) N = 43	ASD Patients (A Group) N = 75	Healthy CHILDREN (HC Group) N = 86
Mean	565	405	403
Standard deviation	163	109	86
Standard error	24	12	9
Minimum	303	223	199
Maximum	986	751	659
Range	683	528	460
CV (coefficient of variation)	0.29	0.27	0.21
Median	534	384	414

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
