# Peer review of "Ribosomal DNA Abundance in the Patient’s Genome as a Feasible Marker in Differential Diagnostics of Autism and Childhood-Onset Schizophrenia"

_jpm, 2022, doi:10.3390/jpm12111796_

Round 1

Reviewer 1 Report

Ershova and colleagues present a study investigating the use of rDNA copy number as a marker for the differential diagnosis of autism and childhood schizophrenia. The manuscript is well written and clear, the methods are appropriate to answers the questions proposed, and the discussion id thorough. And I only have minor suggestions:

line 28: p<10-6

line 88: use higher or large number (lager give the impression of the copies being bigger in size)

line 120: use the full name - Personal and Social Performance (PSP) scale

line 202: I think it should say: "For reference, dashed lines in Fig.1C..." as Fig.1B has no dashed lines

Adjust the placement of the letter D on figure 1, the way it is present is confusing

A sentence or two about the limitations of the technique use for the rDNA copy number assessment should be added, i.e.: limited resolution, age implications, etc.

Author Response

Dear Reviewer,
We highly appreciate your valuable comments.

Comments and Suggestions for Authors

Ershova and colleagues present a study investigating the use of rDNA copy number as a marker for the differential diagnosis of autism and childhood schizophrenia. The manuscript is well written and clear, the methods are appropriate to answers the questions proposed, and the discussion id thorough. And I only have minor suggestions:

line 28: p<10-6

Corrected: p<10^-6

line 88: use higher or large number (lager give the impression of the copies being bigger in size)

Corrected: higher

line 120: use the full name - Personal and Social Performance (PSP) scale

Corrected

line 202: I think it should say: "For reference, dashed lines in Fig.1C..." as Fig.1B has no dashed lines

Corrected

Adjust the placement of the letter D on figure 1, the way it is present is confusing

Corrected

A sentence or two about the limitations of the technique use for the rDNA copy number assessment should be added, i.e.: limited resolution, age implications, etc.

2.5.2. Determining rDNA copy count in DNA samples

Added:

Unlike the generally acknowledged qPCR, NQH technique requires a large amount of DNA  (100 ng) and not applicable to unique (single-copy) genes. However, NQH technique outputs a similar dependence of the signal intensity on DNA damage degree for both tandem rDNA and non-tandem repeats. Previously, it was shown that rDNA is significantly damaged under conditions of oxidative stress [19].

Reviewer 2 Report

Based on previous finding in adults with schizophrenia, Ershova et al. report a higher number of rDNA copies in children (younger than 12) diagnosed with schizophrenia, compared to children with autism spectrum disorder (ASD) and healthy controls. They suggest that this could be a diagnostic marker differentiating between childhood onset schizophrenia (COS) and ASD. Although intriguing and potentially clinically useful, I have major concerns regarding the validity of results. These concerns stem from lack of any assurance that the diagnosis of schizophrenia in the studied children is correct. The authors do state that they followed ICD-10 and DSM-5 criteria, and used PANSS to characterized the status COS patients, still in their clinical description of the group they fail to mention the hallmark symptoms of schizophrenia, namely delusions and hallucinations. Indeed, these two symptoms are the ones that are necessary to differentiate schizophrenia from previously diagnosed ASD according to DSM. In this regard it is difficult to understand how the authors were able to recruit such a large sample of COS patients (N=43), for a very rare disorder with a prevalence of 1:40,000. This is a striking number, even if it took a few years to achieve (the authors do not give details on that), and even when considering the central location of the researchers in a country with a large population. For example, it took the NIMH group (Rapoport et al.) over 20 years to recruit the sample of ~100 COS patients from all over North America. Thus, I have doubts that all of the COS patients really had this rare and difficult to diagnose disorder. Another major concern is the possible effect of medication on rDNA (given that the authors describe other environmental effects on no. of copies, such as age). This by itself can explain the difference between the groups, as the group diagnosed with COS (correct or not) was probably in a more severe state and needed more medications. The highly significant difference (that stems from a subset of COS patients) between the two groups of patients, and the lack of difference from controls, are very surprising. COS and ASD are very difficult to distinguish clinically, and etiologically are probably related (given for example overlapping genetic findings in both). It is hard to believe that such a difference between them in rDNA should be found. The explanation that the authors give for this specific finding in their discussion is not convincing.       

Author Response

Dear Reviewer,

Thank you for your efforts taken to review the manuscript and the scepticism we’ll try to disperse.

We added Table 1 to indicate the differences in clinical picture we used to distinct early onset schizophrenia from ASD.

The Federal Mental Health Center was established decades ago and works in a centralized healthcare system holding a register of mental cases. Given a prevalence of 1:40,000, we should expect 3,500 cases in the whole country and ~400 in the capital region. Thus, N=43 is about 10% of the expected number just within a 200-mile radius, but many patients came from the other regions as well, that is, there is nothing fantastic in this number, even if we diminish the expected numbers to take into account the patient age limitation (under 13 years only). If somebody could recruit 100, why couldn’t we recruit 43?
It should be also noted that the suspected misdiagnosing, i.e. false considering ASD cases as COS patients, would erase the difference between the two groups (ASD and SCZ), while we actually detected it.

The other major concern of yours was the possible effect of medication on rDNA copy number (CN). We currently proceed from the assumption that rDNA CN is a relatively constant genetic trait. It can globally alter only in the conditions of serious genome instability, such as carcinogenesis. Our data you referred that describe certain age changes in the population distribution of the rDNA copy numbers do not show any environmental effects on the individual CN. These findings only suggest different survival rates in the carriers of medium (adaptive) and marginal (disadaptive) rDNA CN. Carriers of both low and high CN die first, therefore the senile population of survivals have the same mean but less variance. That‘s why the effect starts to be visible just from ~75 years and older. Shortly say, no rDNA copies are added or lost with senile age, but the carriers are lost from the population, and the worse evironmental conditions are, the more strict this selection is. But this hypothesis still needs further testing.

Finally, we are very intrigued that your team has an access to 100 COS cases. Could it be interesting for you to test our findings of higher rDNA CN in COS compared to ASD cases? We are currently peer-reviewing a manuscript submitted to PLOS ONE by Japanese researchers, who replicated our findings of increased ribosomal DNA in adult schizophrenia (SCZ) patients on East Asian population, including postmortem brain samples, not only blood cells. Thus, higher rDNA CN in SCZ seems to be a universal phenomenon and highly likely may relate to COS, but not ASD, although, as you concluded, any clear and proven mechanistic explanation for this specific phenomenon is still missing and the existing speculations to explain this fact are not so far convincing.

Reviewer 3 Report

Dear Authors,

The topic of your research article "Ribosomal DNA abundance in the patient's genome as a feasible marker in differential diagnostics of autism and childhood-onset schizophrenia" is relevant and interesting. You have found higher ribosomal repeats (rDNA) copy numbers in childhood-onset schizophrenia than in autistic children and healthy children. The manuscript is rather a short report. The manuscript is generally understandable.

I have some major suggestions for improving the manuscript:

1. The title of Table 1 does not fully reflect the content. Please indicate in the title of Table 1 that these are rDNA copy numbers values.

2. In section 2.3 (Line 117) you state that "The study group included ... age-matched healthy controls (34 healthy children)". But in section 2.1 you indicated that the number of controls (healthy children) was 86 (Line 99). These proposals contradict each other. How many controls were there?

3. Clinical characterization of study participants is insufficient. Please provide a Table with clinical data for schizophrenia patients, ASD patients, and healthy children.

4. In section 2.3, you indicate that the Childhood Autism Rating Scale, PANSS Scale, and Social Performance Scale were used to assess the condition of patients. Does rDNA copy numbers correlate with these scales?

5. In Figure 1D, you label Logistic curves (1) and ROC curves (2) with numbers. This is best indicated by separate letters (D and E), rather than numbers. The numbering is confusing.

6. The significance of the differences (p value) is not indicated in Figure 1B. Please indicate it. This is also not indicated in the text.

7. The results presented in Figure 1C are not discussed in the text, or there is no reference to this figure in the text.

8. There is text in Russian in the Line 201. Please fix this. The manuscript must be written in English only.

9. In Line 202 you indicate that “dashed lines in Fig. 1B display data….”, but in Fig. 1B no dashed lines. Please display dashed lines in Fig. 1B.

10. Line 203 “ill (N=956) adult subjects” – please replace it with “schizophrenia adult subjects (N=956)”.

Good luck with your further research.

Best regards

Author Response

Dear Reviewer,
We highly appreciate your valuable comments.

Comments and Suggestions for Authors

Dear Authors,

The topic of your research article "Ribosomal DNA abundance in the patient's genome as a feasible marker in differential diagnostics of autism and childhood-onset schizophrenia" is relevant and interesting. You have found higher ribosomal repeats (rDNA) copy numbers in childhood-onset schizophrenia than in autistic children and healthy children. The manuscript is rather a short report. The manuscript is generally understandable.

I have some major suggestions for improving the manuscript:

  1. The title of Table 1 does not fully reflect the content. Please indicate in the title of Table 1 that these are rDNA copy numbers values.

The table number is Table 2 in the revised manuscript (since we added another table before), and we corrected the title: Descriptive statistics for the rDNA copy numbers values  in the three groups studied.

  1. In section 2.3 (Line 117) you state that "The study group included ... age-matched healthy controls (34 healthy children)". But in section 2.1 you indicated that the number of controls (healthy children) was 86 (Line 99). These proposals contradict each other. How many controls were there?

There were two research medical centers where the blood was sampled. 34 healthy children were inspected simultaneously with patients in Federal Center of Mental Health during one research episode, and 52 healthy controls were inspected in Research Centre for Medical Genetics. These details are unnecessary for the readers. So we simply replaced 34 for 86.

  1. 3. Clinical characterization of study participants is insufficient. Please provide a Table with clinical data for schizophrenia patients, ASD patients, and healthy children.

  1. In section 2.3, you indicate that the Childhood Autism Rating Scale, PANSS Scale, and Social Performance Scale were used to assess the condition of patients. Does rDNA copy numbers correlate with these scales?

We found no correlation of rDNA copy numbers with the scales. Perhaps, the sample sizes were too little.

  1. In Figure 1D, you label Logistic curves (1) and ROC curves (2) with numbers. This is best indicated by separate letters (D and E), rather than numbers. The numbering is confusing.

Corrected

  1. The significance of the differences (p value) is not indicated in Figure 1B. Please indicate it. This is also not indicated in the text.

The p-values were indicated in Fig. 1С (in the frames). The figures were corrected.

  1. The results presented in Figure 1C are not discussed in the text, or there is no reference to this figure in the text.

The error has been corrected.

  1. There is text in Russian in the Line 201. Please fix this. The manuscript must be written in English only.

The error has been corrected.

  1. In Line 202 you indicate that “dashed lines in Fig. 1B display data….”, but in Fig. 1B no dashed lines. Please display dashed lines in Fig. 1B.

Fig. 1B is Fig. 1С in the revised version.

 Line 203 “ill (N=956) adult subjects” – please replace it with “schizophrenia adult subjects (N=956)”.

The error has been corrected.

Round 2

Reviewer 2 Report

By recommending to reject the original manuscript I meant it could not be improved in a a manner that allows reconsideration. The author's reply does not change my opinion. Also, Table 1 that they are referring to is missing